# Transcriptomic Analysis of Anthocyanin and Carotenoid Biosynthesis in Red and Yellow Fruits of Sweet Cherry (*Prunus avium* L.) during Ripening

Qinghao Wang [1,†], Luyang Jing [1,†], Yue Xu [1,2], Weiwei Zheng [2] and Wangshu Zhang [1,3,*]

1   Ningbo Innovation Center, Zhejiang University, Ningbo Tech University, Ningbo 315100, China; wqh15657811121@163.com (Q.W.); jingluyang@foxmail.com (L.J.); 17858804095@163.com (Y.X.)
2   College of Horticulture Science, Zhejiang A & F University, Hangzhou 311300, China; zhengww@zafu.edu.cn
3   National & Local Joint Engineering Laboratory of Intelligent Food Technology and Equipment, College of Biosystems Engineering and Food Science, Zhejiang University, Hangzhou 310058, China
*   Correspondence: zjuzws@zju.edu.cn
†   These authors contributed equally to this work.

**Abstract:** The diversity of fruit color in sweet cherry (*Prunus avium* L.) has been attributed to the presence of either anthocyanin or carotenoid. We profiled the anthocyanin and carotenoid metabolites to investigate the different pigments and the underlying regulatory mechanisms of differential expression genes (DEGs) between red and yellow fruits of sweet cherry. We profiled two cultivars, 'Jiangnanhong'(JNH, red fruits) and 'Chaoyang'(CY, yellow fruits) to establish their anthocyanin and carotenoid metabolites by LC-MS/MS and transcriptome analysis by RNA-seq to test the difference in gene expression and metabolic substances between the two varieties. Cyanidin-3-*O*-rutinoside was the most different pigment between two cultivars, the content of which in red fruit was significantly higher than in the yellow one during the whole ripening stage (stage 3 and stage 4). The total carotenoid content in the two color types of fruits was close, but the content in yellow fruit was shown to be more stable after harvest. Based on the transcriptome data, the heatmap of selected structural DEGs showed that all of the anthocyanin genes expressed significantly higher levels in red fruits than that in yellow fruits. Two unigenes encoding chalcone synthase (*CHS*) and UDP glucose-flavonoid 3-*O*-glucosyltransferase (*UFGT*) were expressed 1134.58 and 1151.24 times higher in red than in yellow fruits at stage 4, respectively. Correlation analysis showed that anthocyanin genes in JNH were negatively correlated with those in CY; by contrast, there were some strong correlations observed between the two cultivars in carotenoid genes. Thus, the coloration of sweet cherry was mainly attributed to anthocyanin-related genes.

**Keywords:** sweet cherry; RNA-seq; LC-MS/MS; anthocyanin; carotenoid; biosynthesis





## 1. Introduction

Sweet cherry (*Prunus avium* L.) has an attractive appearance, desirable taste, dietary properties and various biological activities, which not only directly influence consumer choice, but are also, today, important directions in the selection and breeding of new fruit varieties selection and breeding [1]. The color of the sweet cherry fruit ranges from dark red to yellow and is determined by the pigments present (e.g., anthocyanins, carotenoids, etc.). Most sweet cherry species with a red color are particularly rich in anthocyanins, and are predominantly cyanidin glycosides [2–4].

Anthocyanin is a class of water-soluble natural pigment in plants that widely presents in nature and belongs to a subgroup of flavonoids. To date, there have been more than 635 anthocyanins identified, approximately 95% of which were derived from six anthocyanins: pelargonidin, cyanidin, delphinidin, peonidin, petunidin, and malvidin [5–7]. Various enzymes have been reported to be involved in the anthocyanin biosynthesis process,

including chalcone synthase (*CHS*), chalcone isomerase (*CHI*), flavanone 3-hydroxylase (*F3H*), flavonoid 3′-hydroxylase (*F3′H*), dihydroflavonol-reductase (*DFR*), and anthocyanidin synthase (*ANS*) [8]. The mechanisms of anthocyanin biosynthesis have been elucidated in a variety of plant species, such as apple, peach, sweet cherry, eggplant, and tomato [9–13]. The *UFGT* family showed differences between red and non-red apple fruits in a previous study. Among them, *UFGT2* was up-regulated only in the non-red cultivar, while *UFGT4* was up-regulated only in the red cultivar, indicating that *UFGT* play an important role in anthocyanin accumulation in apple skins of different colors [9]. In the epidermis of purple eggplant (*Solanum melongena*), MYB86 could directly bind to the promoters of *CHS*, *F3H*, and *ANS* and suppress their activities, resulting in the reduction of the anthocyanin [12]. In sweet cherry, the anthocyanin content increased gradually during the development, especially from the young fruits to the breaker fruits stage. Some unigenes have been reported to be up-regulated in red-colored fruit, including *4CL*, *CHS*, *CHI*, *F3H*, *DFR*, etc. Among them, *UFGT* was suggested to play a key role in red and bicolored (red and yellow) fruits, the expression of which are maintained at high levels in red cultivars in the late ripening stage. Some transcription factor families, such as MYB, bHLH and WD40, may be involved in anthocyanin biosynthesis control [11,14].

Carotenoids are a class of widespread secondary metabolites that are crucial to humans and plants, mainly divided into α-carotene, β-carotene, lycopene, lutein, astaxanthin, and zeaxanthin. Carotenoids are not synthesized by humans, but are essential for several functions, so they must be obtained through the diet, especially from fresh fruits [15,16]. At present, many data are available related to the molecular characterization of genes involved in carotenoid biosynthesis. Hu explained that one *LCYE* gene (LOC113688784) was expressed 2.21 times higher in matured seeds of red coffee (*Coffea arabica*) than that in yellow coffee seeds [17]. Li found that the imbalanced regulation of *HYE* and *HYB* resulted in an increase in zeaxanthin and a decrease in lutein in Chinese raspberry (*Rubus chingii*) [18]. Likewise, Wu reported that the down-regulation of transcription factor *SlMYB72* of R2R3-MYB promoted β-carotene production and decreased lycopene content in tomato (*Solanum lycopersicum*) [19]. Despite these findings, the molecular regulation mechanism of carotenoid biosynthesis between different colored sweet cherry fruits remained unclear.

RNA-seq analysis is commonly used to identify the differential genes and understand their mechanisms underlying gene expression regulation in the biosynthetic pathway of different species from a transcriptomic perspective. Based on the transcriptome analysis, scientists have found that the reduction in key anthocyanin gene expression at the late stage in white bayberry (*Morella rubra*) compared with red bayberry fruits, resulting in a failure to turn red [20]. Wei obtained the transcriptome and DGE profiling data to compare the transcripts involved in the anthocyanin biosynthesis [11]. However, for carotenoid biosynthesis, reports on the transcriptional regulation at specific ripening stages and finally determining the quality in red and yellow sweet cherries are still limited. Herein, in order to identify the major disparity in pigmental substance and the molecular mechanisms in anthocyanin and carotenoid biosynthesis of sweet cherry, we have explored the anthocyanin and carotenoid compounds and transcriptome profiling of a red cultivar (JNH) and a yellow cultivar (CY) in four ripening stages.

## 2. Materials and Methods

### 2.1. Samples

Two cultivars, identified by Zhejiang Provincial Crop Variety Approval Committee (JNH, red fruits)(*Prunus avium* L. cv. 'Jiang Nan Hong') and (CY, yellow fruits)(*Prunus avium* L. cv. Chao Yang) and used for this study, have been successfully cultivated in some regions of southeast China (Figure S1). In March 2021, all the samples of JNH and *CY* were collected from an orchard in Yuyao county, Zhejiang province, China (121°16′ E, 30°1′ N; elevation 6 m). Three individual trees were selected for each cultivar, and the fruits were harvested from each tree at 4 ripening stages: 20 days after flowering (20 DAF, stage 1), 27 DAF (stage 2), 34 DAF (stage 3), and 41 DAF (stage 4) (Table 1, Figure S2A). Additionally,

in order to further explore the changes in pigment contents after harvesting, the fruits of JNH and CY were still collected every 2 days until the color no longer changed, and 5 and 3 maturity levels of two cultivars were obtained (Table 2, Figure S2B). All samples were immediately placed in an ice box and taken back to the laboratory, blended and ground into powder in liquid nitrogen. The RNA-seq samples (JNH and CY fruits in 4 ripening stages) were encapsulated in dry ice and sent to BGI Genomics Co., Ltd. (Wuhan, China) for transcriptome sequencing, and the remaining samples were preserved at −80 °C for subsequent research.

**Table 1.** The cultivars used in the present study.

| Cultivars | Scientific Name | Abbreviation | Ripening Stage [a] | Maturity Level [b] |
|---|---|---|---|---|
| 'Jiangnanhong' | *Prunus avium* | JNH | S1, S2, S3, S4 | I, II, III, IV, V |
| 'Chaoyang' | *P. avium* | CY | S1, S2, S3, S4 | I, II, III |

[a] S1, S2, S3, and S4 refer to different ripening stages; see Figure S2A for detail. [b] I, II, III, IV, and V refer to different maturity levels, see Figure S2B for detail.

**Table 2.** The morphological trait and quality index value of the cultivars' fruit in the fully mature period.

| Cultivars | Color [a] | a/b [b] | Weight (g) | Length (cm) | Width (cm) | TSS (°Brix) [c] | TA (%) [d] |
|---|---|---|---|---|---|---|---|
| JNH | Red | 2.73 ± 0.43 a | 8.73 ± 0.69 a | 21.29 ± 0.88 a | 25.31 ± 1.25 a | 15.77 ± 0.49 a | 0.40 ± 0.02 a |
| CY | Yellow | 0.31 ± 0.01 b | 7.93 ± 0.46 a | 21.56 ± 0.29 a | 26.57 ± 0.37 a | 14.93 ± 1.06 a | 0.43 ± 0.01 a |

[a] see Figure S1 for detail. [b] The ratio of red-green to yellow-blue. [c] TSS, total soluble solid content. [d] TA, titratable acidity. Significant differences were calculated using one-way analysis of variance (ANOVA) followed by Duncan's multiple-range test at the 5% level ($p < 0.05$), and are shown in Tables 2 and 3 with lowercase letters (a, b, c, etc.) between samples.

## 2.2. Determination of Anthocyanin and Carotenoid Content

Anthocyanin content was determined using the method described by Wei [11] with a minor modification. The anthocyanins were extracted in 10 mL methanol (containing 0.05% hydrochloric acid) for 24 h at 4 °C in darkness. The anthocyanin content was measured using the TripleTOF 5600 LC-MS/MS (AB SCIEX, Framingham, MA, USA) at 520 nm. Anthocyanin compound standards, including cyanidin-3-*O*-glucoside (C3G), cyanidin-3-*O*-rutinoside chloride (C3R) and pelargonidin-3-glucoside chloride (P3G), were purchased from Sigma Aldrich Chemical Co. (St. Louis, MO, USA).

The carotenoid content was determined in three replications. A total of 1 g of the sample powder (pre-ground in liquid nitrogen) was homogenized with an extraction solution (5 mL, 80% pre-cooled acetone) and incubated overnight at 4 °C in the dark. Next, the samples were centrifuged at 7500 rpm for 20 min at 4 °C and absorbance was measured spectrometrically at 470 nm (80% acetone was used as a control group). Carotenoid content was calculated according to Dere, Günes, and Sivaci, and expressed as μg/g fresh weight [21].

## 2.3. Total RNA Extraction

Total RNA was extracted using the polysaccharide polyphenol reagent kit (Takara Cat. 9769, Beijing, China). The integrity of RNA was determined by 1.5% agarose gel electrophoresis (Figure S3). The purity and concentration of RNA were determined by Nandrop 2000C (Thermo, Waltham, MA, USA).

## 2.4. Library Construction and Transcriptome Sequencing

Oligo (dT)-attached magnetic beads were used to purify mRNA, which was fragmented into small pieces with a fragment buffer at appropriate temperature. Then, first-strand cDNA was generated using random hexamer-primed reverse transcription, followed by a second-strand cDNA synthesis. Afterwards, A-tailing mix and RNA index adapters

were added by incubating to end repair. The cDNA fragments obtained from the previous step were amplified by PCR, and products were purified by Ampure XP beads then dissolved in EB solution. The product was validated on the Agilent Technologies 2100 bioanalyzer for quality control. The double stranded PCR products were heated, denatured and circularized by the splint oligo sequence to obtain the final library. The single strand circle DNA (ssCir DNA) was formatted as the final library. The final library was amplified with Phi29 polymerase to make DNA nanoball (DNB), which had more than 300 copies of one molecula. DNBs were loaded into the patterned nanoarray, and pair end 100 bases reads were generated on the BGIseq500 platform (BGI, Shenzhen, China). This project sequencing used the DNBSEQ platform and referenced an existing genome (https://www.ncbi.nlm.nih.gov/nuccore/NW_018922178.1) (accessed on 12 June 2017) from NCBI. The sequencing raw data are deposited in the NCBI Sequence Read Archive under BioProject ID (Accession No. PRJNA905870).

### 2.5. Sequencing Data Filtration and Transcripts Mapping

The sequencing data were filtered with SOAPnuke (v1.5.2) by (1) removing reads containing a sequencing adapter; (2) removing reads whose low-quality base ratio (base quality less than or equal to 5) was more than 20%; (3) removing reads whose unknown base ('N' base) ratio was more than 5% [22]. Afterwards, clean reads were obtained and stored in FASTQ format and then mapped to the reference genome (https://www.ncbi.nlm.nih.gov/nuccore/NW_018922178.1) (accessed on 12 June 2017) using HISAT2 (v2.0.4) [23]. BOWTIE (v2.2.5) was applied to align the clean reads to the gene set, followed by de novo transcript prediction and differentially spliced gene detection [24].

### 2.6. Functional Gene Annotation

The ORF of unigene was detected by GETORF (EMBOSS 6.5.7) package, and the ORF was mapped to the structural domain of transcription factor protein using HMM-SEARCH 3.0 package (http://www.ebi.ac.uk/Tools/hmmer) (accessed on 12 June 2017). The Kyoto Encyclopedia of Genes and Genomes (KEGG) database categories were assigned to the unigene sequences using the KEGG Automatic Annotation Server (KAAS) online. Blast2GO v2.5 was used to obtain the Gene Ontology (GO) annotation of unigenes. The gene expression levels were calculated by RSEM (v1.2.12).

### 2.7. Different Expression Gene (DEG) Analysis

Differential expression analysis was performed using the DESeq R (v1.4.5) package. To gain insight to the change in phenotype, the Kyoto Encyclopedia of Genes and Genomes (KEGG) and Gene Ontology (GO) enrichment analysis of annotated different expression genes were performed by Phyper function based on a hypergeometric test (https://en.wikipedia.org/wiki/Hypergeometric_distribution) (accessed on 12 June 2017). The significant levels of terms and pathways were corrected by Q-value with a rigorous threshold (Q value $\leq$ 0.05) using the Bonferroni method.

### 2.8. Quantitative Real-Time Reverse Transcription PCR (qRT-PCR) Analysis

In order to verify the expression pattern results determined by RNA-seq, six and four unigenes involved in anthocyanin and carotenoid metabolic ways were chosen for qRT-PCR analysis, respectively. Total RNA was extracted by the same method mentioned above. The specific primer pairs and reference gene (*β-Actin*) are shown in Table S4. cDNA was obtained using the ABScript II RT Mix reverse transcription kit (ABclonal Co., Ltd., Wuhan, China). qRT-PCR analysis was also performed using a kit, according to protocol (Vazyme biotech Co., Ltd., Nanjing, China). The following reactions were performed using a real-time PCR instrument (ABI Q6 Flex) with a procedure as follows: 95 °C,1 min; 95 °C 15 s; 60 °C 1 min (45 cycles of the above steps). Each plate was repeated three times in independent runs for all reference and selected genes. The real-time data were analyzed by CFX ManagerTM 3.0 software to obtain amplification and melting curves,

which were transformed to analyze the data results using the $2^{-\Delta\Delta CT}$ method. Three biological replicates and three technical replicates were performed for each sample.

*2.9. Statistical Analysis*

The content of each chemical compound was shown as the mean ± standard deviation of three replicates. Statistical analysis was performed using SPSS 19.0 software (SPSS Inc., Chicago, IL, USA). Significant differences were calculated using one-way analysis of variance (ANOVA) followed by Duncan's multiple-range test at the 5% level ($p < 0.05$), and are shown in Tables 2 and 3 with lowercase letters (a, b, c, etc.) between samples. Undetectable substances are marked with "nd" in all chemical profile tables. Correlation analysis was carried out by Pearson's test ($p < 0.05$), and the correlation figure was drawn with Origin 8.0.

**Table 3.** Anthocyanin and carotenoid content in the fruits of cultivars in different developmental stages.

| | Sample | Cyanidin-3-*O*-Rutinoside (µg/g·FW) | Cyanidin-3-*O*-Glucoside (µg/g·FW) | Pelargonidin 3-Glucoside (µg/g·FW) | Total Carotenoid (µg/g·FW) |
|---|---|---|---|---|---|
| Ripening stages | JNH S1 | 0.23 ± 0.01 d | 4.35 ± 0.35 a | nd | 2.03 ± 0.05 f |
| | JNH S2 | 2.62 ± 0.18 c | 1.46 ± 0.06 c | nd | 4.22 ± 0.34 e |
| | JNH S3 | 38.04 ± 2.31 b | 1.10 ± 0.05 d | nd | 11.78 ± 0.69 b |
| | JNH S4 | 719.31 ± 19.84 a | 1.93 ± 0.06 b | 0.13 ± 0.01 a | 11.17 ± 0.86 c |
| | CY S1 | nd | 0.19 ± 0.03 e | nd | 2.48 ± 0.14 f |
| | CY S2 | nd | 0.17 ± 0.07 e | nd | 7.05 ± 0. 07 d |
| | CY S3 | 0.32 ± 0.01 e | 0.12 ± 0.02 e | nd | 11.79 ± 0.28 b |
| | CY S4 | 2.34 ± 0.21 c | 0.09 ± 0.02 e | nd | 12.86 ± 0.38 a |
| Maturity level | JNH I | 112.62 ± 5.59 d | 1.00 ± 0.06 d | nd | 10.63 ± 0.02 d |
| | JNH II | 111.07 ± 5.17 d | 1.13 ± 0.04 d | nd | 9.01 ± 0.10 c |
| | JNH III | 837.70 ± 15.12 c | 1.71 ± 0.02 c | nd | 8.27 ± 0.07 b |
| | JNH IV | 1424.16 ± 7.56 b | 4.29 ± 0.05 b | nd | 6.53 ± 0.09 e |
| | JNH V | 3909.32 ± 112.67 a | 13.54 ± 0.28 a | 0.23 ± 0.01 a | 3.19 ± 0.21 f |
| | CY I | 1.35 ± 0.23 e | 0.94 ± 0.03 de | nd | 13.49 ± 0.06 a |
| | CY II | 1.44 ± 0.12 e | 1.07 ± 0.16 d | nd | 13.75 ± 0.08 a |
| | CY III | 6.25 ± 0.44 e | 0.74 ± 0.11 e | nd | 13.80 ± 0.13 a |

[a] Data expressed as means ± standard deviation of samples' fresh weight in different ripening stages and maturity levels. [b] nd, the substance was not detectable. Significant differences were calculated using one-way analysis of variance (ANOVA) followed by Duncan's multiple-range test at the 5% level ($p < 0.05$), and are shown in Tables 2 and 3 with lowercase letters (a, b, c, d, e, f) between samples.

## 3. Results

*3.1. Anthocyanin and Carotenoid Content at Different Ripening Stages and Maturity Levels of Sweet Cherry*

Three types of anthocyanins, cyanidin-3-*O*-glucoside (C3G), cyanidin-3-*O*-rutinoside chloride (C3R) and pelargonidin-3-glucoside chloride (P3G), and total carotenoid were identified (Table 3). Among them, C3R was the most abundant anthocyanin in both fruits of JNH and of CY, but P3G was almost undetectable. Notably, C3R significantly increased after S3 in JNH, and continued to increase rapidly until postharvest. However, in CY, C3R content was always kept at a low level. The content of C3R reached a maximum of 3909.32 µg/g at JNH V, but only reached a maximum of 6.25 µg/g in CY III, which was only 0.16% of that in JNH V. Overall, the anthocyanin content presented a prominently higher level in red cherries (JNH) than that in yellow cherries (CY) during the ripening stages and after harvest in this study.

With regard to the total carotenoid, its content was much less than the anthocyanin in two cultivars. The amount of total carotenoid content increased gradually during ripening in JNH and CY. However, after maturity, the carotenoid content in CY III (13.80 ug/g) was 4.33 times higher than that in JNH V (3.19 ug/g). Moreover, an obviously downward trend was observed in the carotenoid content of JNH from maturity level I to V, whereas the content change in CY appeared more stable (Table 3).

### 3.2. Transcriptome Profile of the RNA Libraries of Sweet Cherry

We constructed eight cDNA libraries from the total RNA of JNH_S1, JNH_S2, JNH_S3, JNH_S4, CY_S1, CY_S2, CY_S3, and CY_S4. These cDNA libraries were subjected to pair-end reading with the BGIseq500 platform (BGI, Beijing, China). This RNA-Seq project referenced an existing genome (https://www.ncbi.nlm.nih.gov/nuccore/NW_018922178.1) (accessed on 12 June 2017) from NCBI (Table S1).

The preprocessing result of sequencing data is shown in Table 4. It can be seen that after the quality pretreatment of the original sequencing sequence, the minimum percentage of clean reads in raw reads was 93.20%, the minimum of Q30 was 90.94%, and the minimum mapping rate was 86.74%. The assembly statistics are shown in Table S2. The sequencing and assembly results suggest that the unigene data were highly reliable for further analysis.

**Table 4.** The quality assessment of sequencing data of cultivars with different packaging materials.

| Sample | Clean Bases (Gb) | Clean Reads/Raw Reads (%) | Q20 (%) | Q30 (%) | Total Mapping (%) |
|---|---|---|---|---|---|
| JNH S1 | 6.38 | 97.03% | 97.44 | 92.21 | 86.76 |
| JNH S2 | 6.44 | 97.95% | 97.32 | 91.79 | 88.95 |
| JNH S3 | 6.43 | 97.79% | 97.05 | 90.94 | 88.34 |
| JNH S4 | 6.41 | 97.49% | 97.43 | 92.01 | 87.60 |
| CY S1 | 6.40 | 93.88% | 97.87 | 93.68 | 91.12 |
| CY S2 | 6.38 | 93.60% | 97.91 | 93.83 | 91.71 |
| CY S3 | 6.35 | 93.20% | 97.96 | 93.93 | 90.75 |
| CY S4 | 6.39 | 93.73% | 97.92 | 93.82 | 90.50 |

### 3.3. Functional Annotation and Classification

For the functional annotation of the sweet cherry transcriptome, 26,247 unigenes and 14,523 transcripts were aligned with sequences from the reference genome (Table S1). Of the 26,247 unigenes, about 23,308 (88.80%) unigenes showed homology to the sequences in the reference genome. Additionally, 2939 (11.20%) novel genes were identified in this study. In order to identify new transcript regions, we compared the assembled transcripts with those annotated with the reference sequence, and a total of 3014 (20.76%) novel transcripts were identified. The novel transcripts must meet the following conditions: 200 bp or more from the existing annotated transcript, and not shorter than 180 bp in length (Table 5).

**Table 5.** Summary of the annotations for the sweet cherry unigenes and transcripts.

| Items | Number of the Unigenes and Transcripts | Percentage of Annotated Unigenes and Transcripts (%) |
|---|---|---|
| Total genes | 26,247 | 100 |
| Known genes | 23,308 | 88.80 |
| Novel genes | 2939 | 11.20 |
| Total transcripts | 14,523 | 100 |
| Known transcript | 11,509 | 79.25 |
| Novel transcript | 3014 | 20.75 |

Gene Ontology (GO) enrichment analysis was used to classify the function of the DEGs, including biological process (BP), cellular component (CC), and molecular function (MF). In the BP category, a large number of genes were distributed in the cellular process (7388, 11.83%) and metabolic process (5797, 9.28%). As for the CC category, the majority of unigenes were grouped into the cellular anatomical entity (11356, 18.18%) and intracellular category (6583, 10.54%). Genes in the MF category were primarily sorted into the binding (8997, 14.41%) and catalytic activity (8630, 13.82%) categories (Figure 1A). The DEGs-enriched KEGG pathway scatter plot of JNH vs. CY at four different ripening stages is shown in Figure 1B.

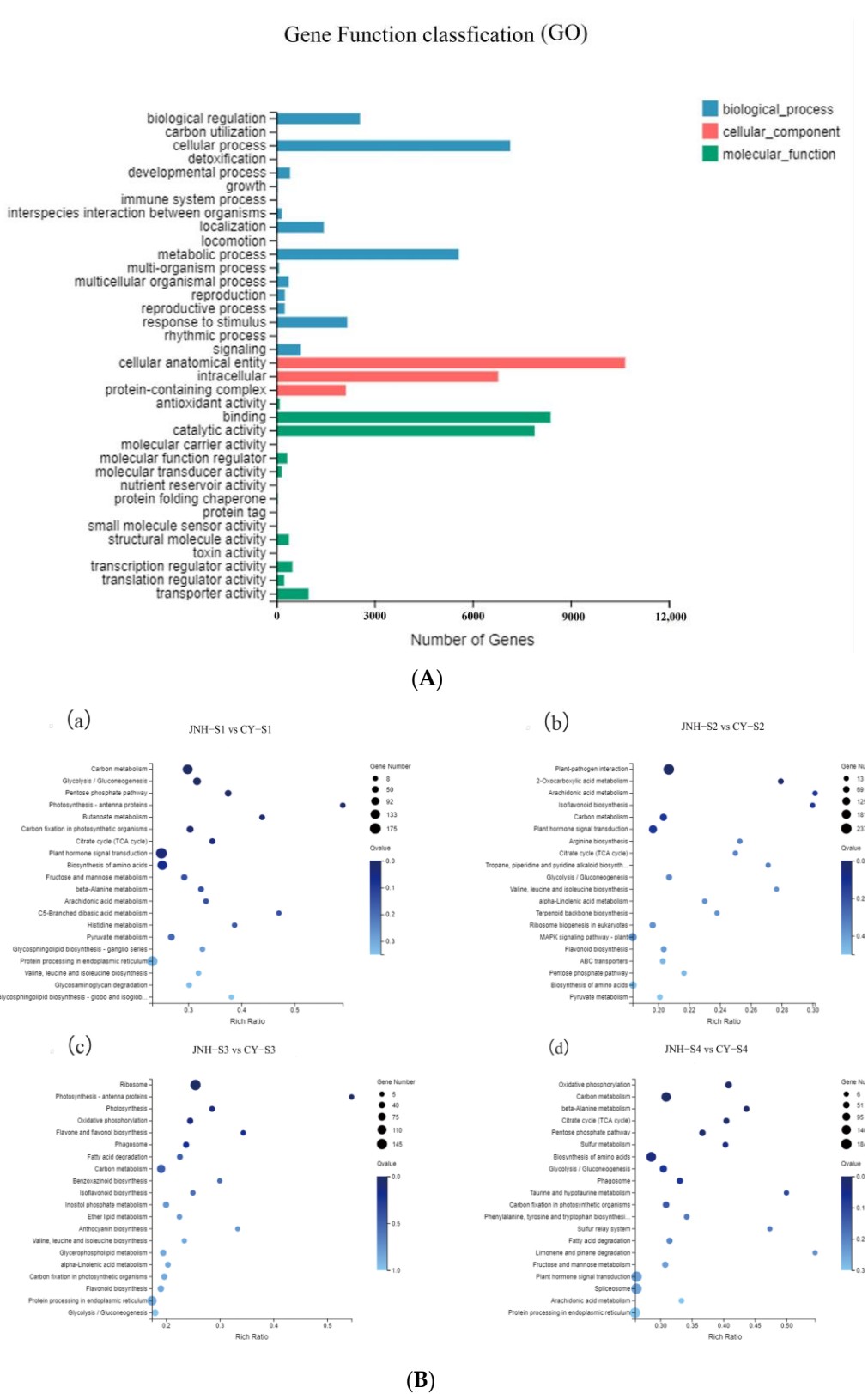

**Figure 1.** GO classification of unigenes (**A**) and DEGs-enriched KEGG pathway scatter plot (**B**) of red and yellow fruits of sweet cherry in four ripening stages. Pathway names are listed along the vertical axis, and the horizontal axis represents enriched factors that correspond to the pathway. (**a**) is the S1 period, (**b**) is the S2 period, (**c**) is the S3 period, (**d**) is the S4 period.

### 3.4. Differential Genes Expression

In this study, we screened the differentially expressed genes (DEGs) under the criteria of $|\log2FC| \geq 1$, and FDR < 0.05. We compared the DEGs within a cultivar and between two cultivars within a specific stage, and the number of up- and down-regulated unigenes is shown in Figure 2A. Regarding the fruit ripening, the number of the up- and down-regulated genes between JNH and CY decreased and then increased from stage 1 to stage 4. Of these, ripening stage 4 possessed the largest number of DEGs: approximately 3409 up-regulated and 3222 down-regulated unigenes between JNH4 and CY4. In addition, within the individual cultivars, we found that the number of DEGs between the ripening stages 1 and 4 were significantly higher than any other comparisons.

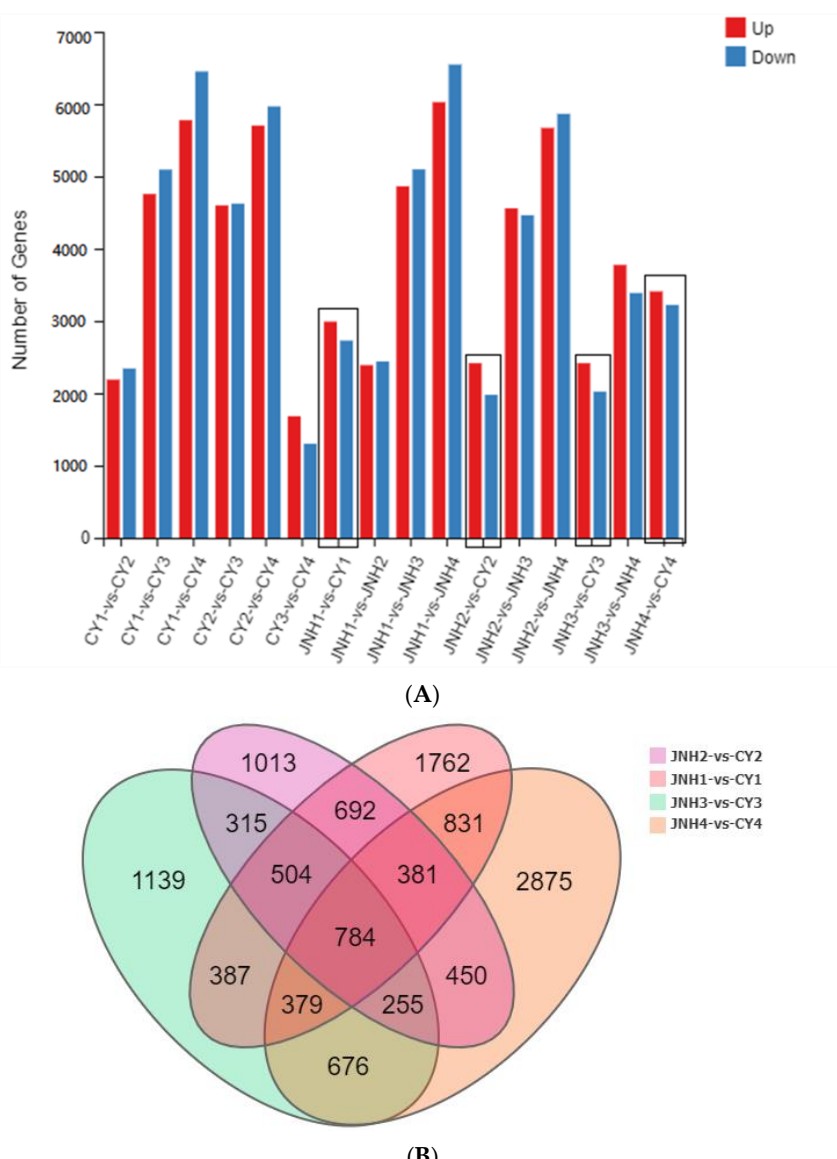

**Figure 2.** Overview of the differentially expressed genes (DEGs) of JNH and CY fruits at four different ripening stages. (**A**). Number of DEGs between different stages of fruit and between cultivars identified by pairwise comparison (log2 FC ≥ 1 and adjusted FDR < 0.05). The bar chart framed in black boxes were the DEGs between JNH and CY in four specific ripening stages. (**B**). A Venn diagram showing the numbers of the DEGs in four pairwise comparisons.

The DEGs were visualized in Venn diagrams to observe the overlap of expression patterns. A total of 12,443 DEGs between the JNH and CY in four different ripening stages

were identified, of which 784 (6.30%) genes showed significant differential expression in all the four pairwise comparisons (Figure 2B).

### 3.5. Identification of DEGs Involved in Anthocyanin and Carotenoid Metabolic Pathways

We analyzed the expressions of eleven and six structural genes involved in anthocyanin and carotenoid metabolic pathways using RNA-seq (Figure 3). All of the anthocyanin genes (including *PAL*, *C4H*, *CHS*, *CHI*, *F3H*, *F3'H*, *DFR*, *ANS* and *UFGT*) were significantly up-regulated in red fruit JNH (ripening stage 3 and stage 4) in the present study. Furthermore, the expressions of anthocyanin genes in JNH began to significantly increase when fruits began turning red (ripening stage 3). However, all of the DEGs encoding anthocyanin biosynthesis in CY showed significantly lower levels when fruits turned yellow (ripening stage 3 and 4). Based on the transcriptome data, *CHS* and *UFGT* expressed 1134.58 and 1151.24 times higher levels in JNH compared with CY at stage 4, respectively.

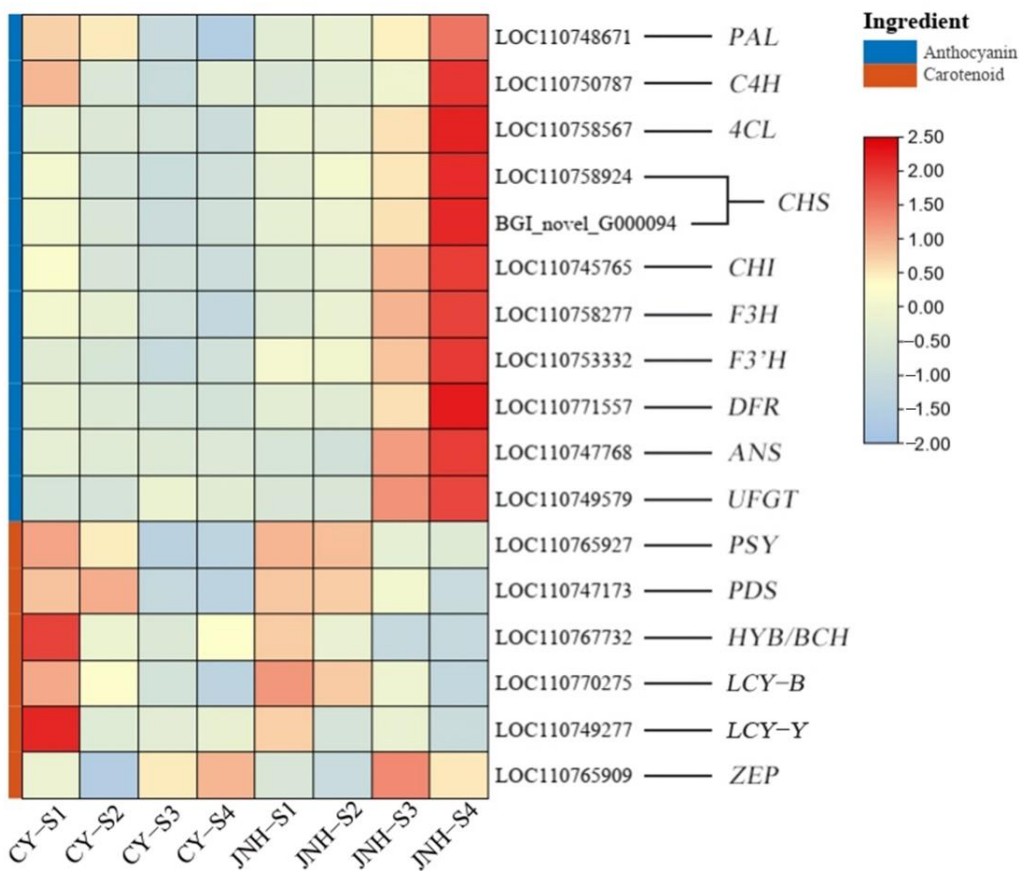

**Figure 3.** Visualizing heatmap of structural gene expression of anthocyanin and carotenoid in the fruit of sweet cherry. S1, S2, S3, and S4 refer to the four different ripening stages of JNH and CY. The underlying transcriptome mapping data are presented in Table S3.

On the other hand, genes (including *PSY*, *PDS*, *HYB*, *LCYB*, and *LCYE*) on the pathway of carotenoid biosynthesis expressed higher levels at stage 1 in two cultivars. Among them, *HYB* and *LCYE* expressed significantly higher levels in CY than that in JNH at stage 1. Interestingly, after stage 1, there was no marked upregulation in the expressions of carotenoid genes (except for *ZEP*) of JNH and CY with ripening. *ZEP* expression was slightly upregulated at stage 3 and stage 4 in both the two cultivars.

In addition, a novel *CHS* gene (BGI_ novel_G000094) was obtained by RNA-Seq in the present study (Figure 3), and its expression was basically consistent with that of *CHS* (LOC 110758924) in the reference genome. The FPKM values underlying the heatmap from the transcriptome mapping data were shown in Table S3.

### 3.6. Correlation Analysis of the DEGs between Red and Yellow Sweet Cherries

To further explore the gene expression relationship between red (JNH) and yellow (CY) cherry fruits in anthocyanin and carotenoid biosynthesis, the correlation analysis was conducted. The correlation coefficient (r) ranged between −1 and 1 (Figure 4). As shown in Figure 4A, except for *UFGT*, all of the anthocyanin genes in CY were negatively correlated with those in JNH. Contrastingly, some strong correlations were observed between JNH and CY fruits in carotenoid genes' expressions. For example, the expression of *PDS* in CY was significantly related to the expression of *PSY* in JNH during the ripening stages (r = 0.99, *p* < 0.05). However, we also found that the expression of *ZEP* in CY was basically negatively correlated with all of the carotenoid genes (except *ZEP*) in JNH (Figure 4B).

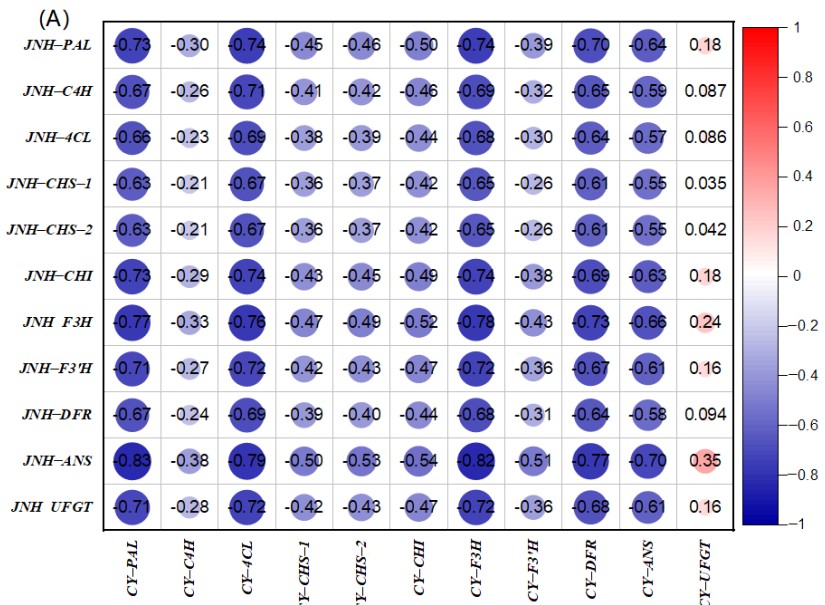

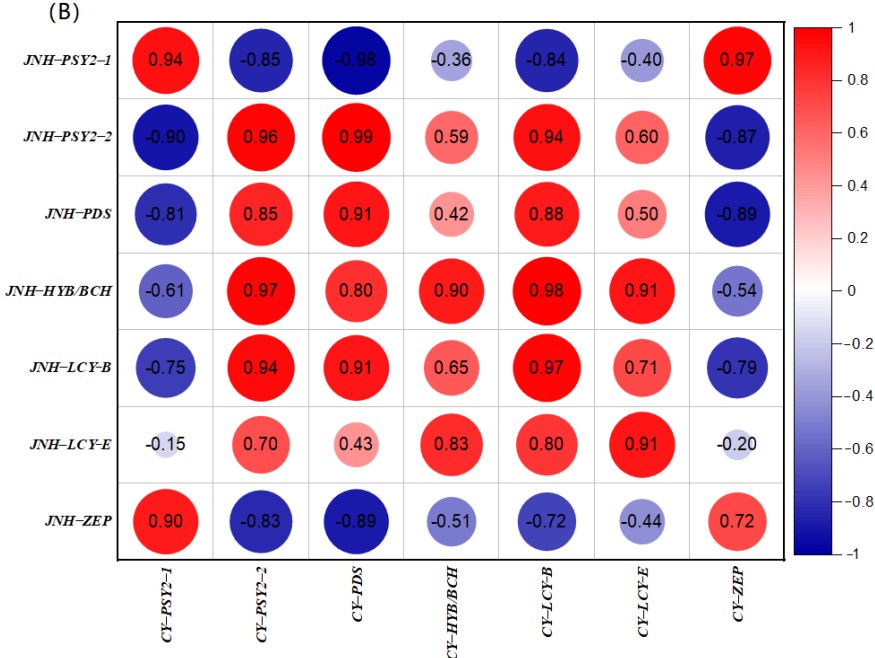

**Figure 4.** Correlation analysis of anthocyanin (**A**) and carotenoid (**B**) genes between JNH and CY fruits from the RNA-seq data. Red indicates positive correlation and blue indicates negative correlation (*p* < 0.05).

### 3.7. Validation of RNA-seq Data Using qRT-PCR

To validate the accuracy of the RNA-seq results, we verified the expression pattern of the 11 genes (*CHS, CHI, F3H, F3′H, DFR, ANS, UFGT, PDS, HYB, LCYB,* and *LCYE)* in red (JNH) and yellow (CY) cherry fruits during the ripening (Figure 5). All of the selected anthocyanin unigenes showed much higher gene expression levels in JNH than CY at ripening stage 3 and stage 4. Additionally, carotenoid structural genes generally expressed lower levels than anthocyanin structural gene expressions in two cultivars. Among them, *HYB* and *LCYE* showed much higher levels in CY compared with JNH at ripening stage 1, which was consistent with the transcriptome data (Figure 3). Furthermore, the correlation coefficient (R) of the expression levels of the 11 genes (*CHS, CHI, F3H, F3′H, DFR, ANS, UFGT, PDS, HYB, LCYB,* and *LCYE*) measured using RNA-seq and qRT-PCR were 0.999, 0.984, 0.993, 0.957, 0.999, 0.997, 0.999, 0.644, 0.803, and 0.964, 0.841 (Figure S4). Thus, the RNA-seq data here are credible.

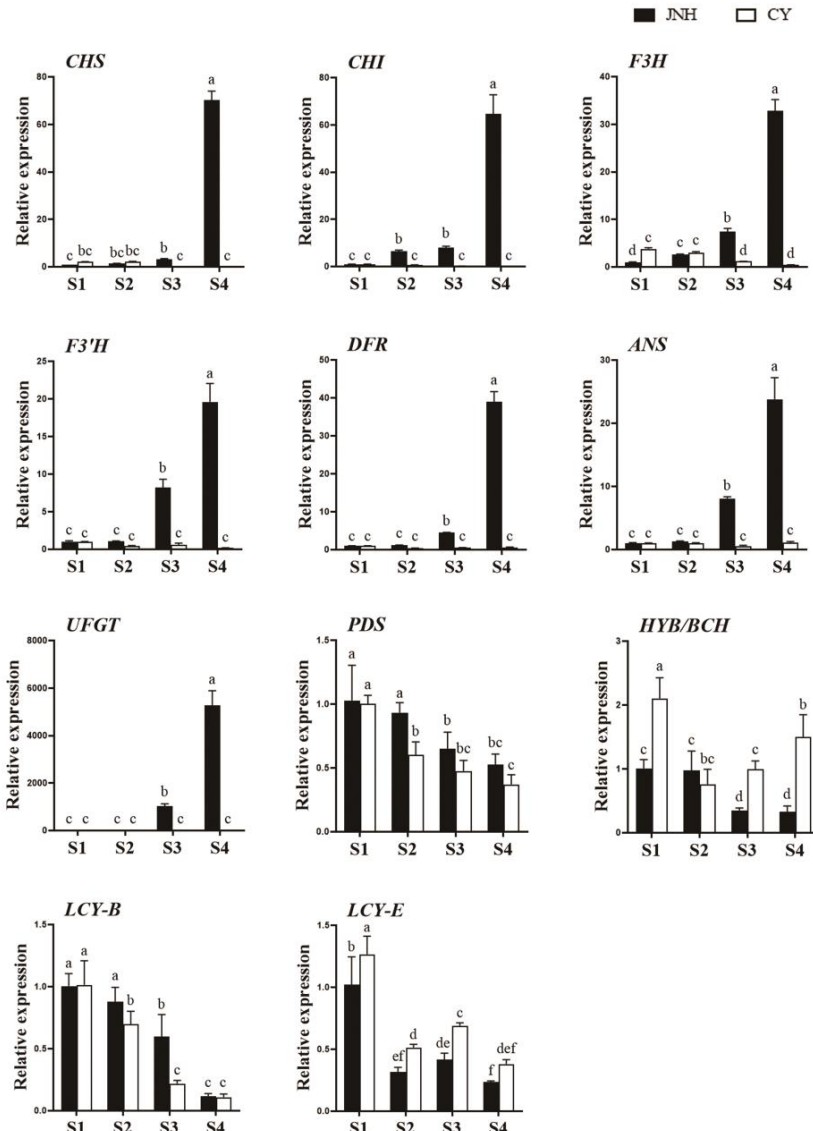

**Figure 5.** Expression analysis of eleven genes related to anthocyanin and carotenoid biosynthesis in sweet cherry using qRT-PCR. The standard error of the mean for three biological replicates (nested with three technical replicates) is represented by the error bars ($p < 0.05$). Significant differences were calculated using one-way analysis of variance (ANOVA) followed by Duncan's multiple-range test at the 5% level ($p < 0.05$), and are shown in Tables 2 and 3 with lowercase letters (a, b, c, d, e, f) between samples.

## 4. Discussion

The development cycle of sweet cherry fruit is short and the color changes quickly. Previous studies have shown that the red color of sweet cherry fruit is mainly due to the accumulation of anthocyanin, in which cyanidin-3-*O*-rutinoside (C3R) is the most abundant compound [25–27]. Our results revealed that C3R, as the predominant anthocyanin in red cherries otherwise, is significantly lower in yellow cherries. It is indicated that C3R plays a pivotal role in fruit coloration in red fruit, but it's not the key role in yellow fruits, especially at ripening stage 4 (Table 3). Additionally, we analyzed the total carotenoid content in red and yellow cherries at four ripening stages and after maturity. The results showed that the content between two cultivars was relatively close during the ripening stages. However, after maturity, the carotenoid content in yellow fruits appeared more stable than that in red fruits. We speculated that this may be due to the down-regulation of anthocyanin genes in yellow cherries after maturity, or it may be because the genes or transcription factors involved in carotenoid biosynthesis were expressed or regulated more stably in yellow cherries, thereby delaying the degradation of carotenoid in CY. Recent studies have also shown that the lack of anthocyanins is one of the reasons that affect the yellowing of sweet cherry fruits [28]. Together, our results suggested that yellow color formation in cherry fruits resulted from insufficient anthocyanin biosynthesis during ripening [29], but did not result from the adequate accumulation of carotenoid.

Previous studies have proposed that *CHS*, *F3H*, *DFR*, and *ANS* increased gradually with fruit ripening in red cherries, but these genes showed significantly lower expressions in half red and half yellow cherries and always kept at a low level [26,30]. Our results showed that these genes expressed significantly higher levels in red fruits than that in yellow fruits at late ripening stages of sweet cherry (Figure 6, Table S3). However, Liu reported that *UFGT* was significantly correlated with the bicolored cultivar 'Caihong', and *CHS* was highly correlated with the red cultivar 'Hongdeng' using principal component analysis (PCA) [14]. These findings were inconsistent with our present results. Our results showed that *CHS* and *UFGT* expressed 1134.58 and 1151.24 times higher levels in red fruit compared with yellow fruit at stage 4, respectively, indicating that higher expression levels of *CHS* and *UFGT* were tightly linked to the sufficient anthocyanin biosynthesis in the fruits of red cherries, but not to yellow cherries. *CHS* is the first key enzyme in the biosynthesis of flavonoids, and it catalyzes the synthesis of naringenin chalcone from three molecules of 4-malonyl-CoA, and one molecule of p-coumaroyl CoA at the beginning of anthocyanin synthesis [31,32]. *UFGT* is a key enzyme that catalyzes the transfer of glycosyl from uridine 5′-diphosphoglucose to anthocyanin 3′-O-glucosyltransferase at the last step of anthocyanin biosynthesis, which is to convert unstable cyanidin into stable anthocyanin [33,34]. Studies have reported that *UFGT* is closely related to anthocyanin biosynthesis in the developmental stages of red apple, red sand pear and grapefruit [35–37]. From our results, we know that *UFGT* is one of the structural genes on anthocyanin biosynthetic pathway. The joint role of UFGTs and other downstream genes in the final stabilization of anthocyanins was as the most differentially expressed gene between red and yellow varieties, which was suggested to be the key gene causing color differences between red and yellow fruits in mature cherry fruits. This may be because *UFGT* could catalyze the attachment of sugar to the anthocyanin aglycone more quickly and more significantly in red fruits compared with yellow fruits, considerably increasing its stability.

Meanwhile, a previous study has shown some regulatory unigenes in the anthocyanin biosynthesis of the red and yellow fruits of sweet cherry, but not in the carotenoid biosynthesis [11]. Here, we analyzed anthocyanin and carotenoid biosynthetic gene expression levels based on the pigmental substances and different ripening stages in sweet cherry fruits. We found that all the structural genes involved in carotenoid biosynthesis showed positive correlations between JNH and CY, except *ZEP* (Figure 6B). This indicated that the expression pattern of carotenoid biosynthetic genes in JNH and CY were relatively similar during the ripening. *ZEP* is an enzyme that catalyzes zeaxanthin to form antherxanthin, and then catalyzes antherxanthin to form violaxanthin. The expression of the ZEP gene

can affect plant photosynthesis, which may affect plant color [38], which was slightly up-regulated in two cultivars at S3 and S4 in the present study, suggesting that the violax-anthin content in red and yellow cherries would increase, relatively, at the late ripening stage. More importantly, β-carotene hydroxylaseas (HYB) was a rate limiting enzyme that catalyzed the formation of zeaxanthin [39,40]. *HYB* was the unigene that encoded HYB formation. *HYB* peaked at the highest expression level at S1 and then decreased in late ripening stages in both of red and yellow cherries [41]. In yellow cherries, in particular, the expression of *HYB* decreased by 65.47% from S1 to S4 (Figure 3, Table S3). On the one hand, this could be due to the interactions of other genes that inhibited *HYB* expression at the late ripening stage; on the other hand, the insufficient catalytic substrate upstream of *HYB* could be another reason for it. Additionally, in yellow cherry fruits, *HYB* expressed a 1.66 times higher level compared with red fruit at S1; we supposed that *HYB* may enhance a flux from β, β-ring hydroxylation to β, ε-ring hydroxylation, which may result in an increase of zeaxanthin in red fruits at early stage of ripening (Figure 6). For this result, the changes in specific carotenoid components need to be further explored. In summary, both anthocyanins and carotenoids can regulate the color of plant fruits [12]. In summary, the imbalanced regulation of *HYB* and *ZEP* and gene-interactions could be the reason for the different color formation in red and yellow cherry fruits in this study.

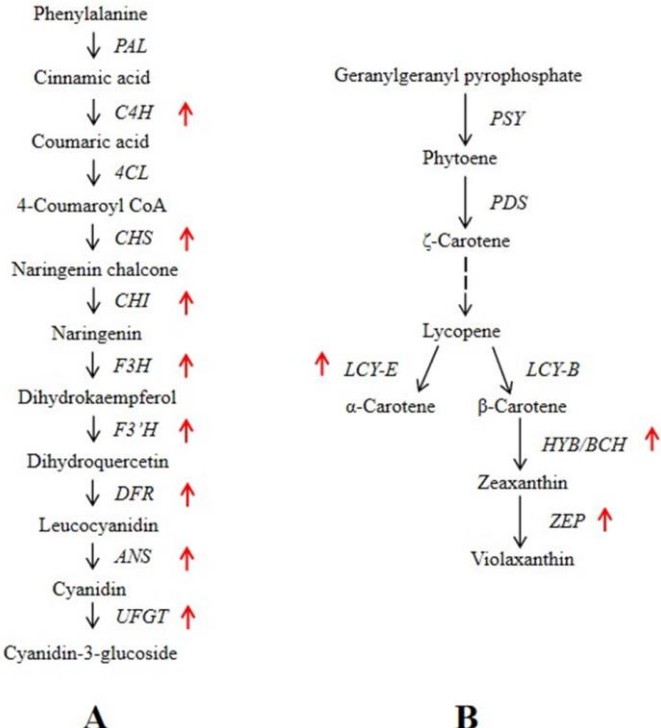

**Figure 6.** Relative pathway and enzymes involved in anthocyanin (**A**) and carotenoid (**B**) biosynthesis of *P. avium* fruit. The up arrow in the anthocyanin pathway indicates JNH S4 vs. CY S4 upregulated. The up arrow in the carotenoid pathway indicates CY S1 vs. JNH S1 upregulated.

The findings presented in this study offer valuable insights into the transcriptional regulation of color biosynthesis during ripening among the different *P. avium* varieties. The discrepant pigmental substance of the color change in red and yellow fruits during ripening stages were identified. It is also recommended to conduct the more metabolome profiling (including primary and secondary substances) among the different varieties to systematically evaluate the metabolite differences for fruit coloration for possible functional biomarker development.

## 5. Conclusions

We have performed a combined analysis of the transcriptome and metabolome of the red (JNH) and yellow (CY) sweet cherry at the four ripening stages. The profiles of anthocyanin and total carotenoid derived from the red and yellow fruits were identified, and the differential expression genes (DEGs) on the anthocyanin and carotenoid biosynthetic pathways were systematically compared. As for the pigmental metabolites, cyanidin-3-*O*-rutinoside (C3R) was the most differing pigment between red and yellow cherry fruits, with a significantly higher content in red fruits than that in yellow fruits; the total carotenoid contents between two cultivars were relatively close in value, but the contents in yellow fruits were shown to be more stable than that in red fruits after maturity. The expressions of *CHS* and *UFGT* in the red fruits were significantly higher than those of yellow fruits at late stages of ripening, but the expressions of structural genes in the carotenoid biosynthetic pathway showed little difference between red and yellow cherries.

**Supplementary Materials:** The following supporting information can be downloaded at: https://www.mdpi.com/article/10.3390/horticulturae9040516/s1, Figure S1: Fruit morphology of JNH and CY are shown. The bars of the sections are 1.0 cm; Figure S2: Different ripening stages (A) and maturity level (B) of JNH and CY were showed. The bars of the sections are both 1.0 cm; Figure S3: The agarose gel electrophoresis (1.5%) testing of the integrity of RNA; Figure S4: Correlation (R2) of the expression levels of the 11 genes (CHS, CHI, F3H, F3′H, DFR, ANS, UFGT, PDS, HYB, LCYB, LCYE) measured by qRT-PCR and RNA-seq; Table S1: Basic information of reference genome; Table S2: The assembly statistic of the RNA-seq in this study; Table S3: The FPKM values underlying the heatmap from the transcriptome mapping data; Table S4: Specific primers pairs for selected genes used in qRT-PCR.

**Author Contributions:** W.Z. (Wangshu Zhang) designed this work and substantively revised the manuscript. Q.W. drafted the manuscript and contributed to the data curation. L.J. and Y.X. performed the experiments and analyzed the data. W.Z. (Weiwei Zheng) contributed to the technical support. All authors have read and agreed to the published version of the manuscript.

**Funding:** This work was supported by Ningbo Science and Technology Planning project (2019B10024).

**Data Availability Statement:** All data generated or analyzed during this study are included in the supplementary information files. The raw RNA-seq data are freely available at https://www.ncbi.nlm.nih.gov/sra/PRJNA905870 (accessed on 27 November 2022). There are no ethical issues involved in this work. Experimentation with living animals did not happen in this work.

**Conflicts of Interest:** The authors declare no conflict of interest.

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
