# Peer review of "Transcriptomic Analysis of Anthocyanin and Carotenoid Biosynthesis in Red and Yellow Fruits of Sweet Cherry (Prunus avium L.) during Ripening"

_horticulturae, doi:10.3390/horticulturae9040516_

Round 1

Reviewer 1 Report

The authors present a systematic study on the profiles of anthocyanin and total carotenoid derived from red and yellow sweet cherries at four ripening stages. The biosynthetic pathways of anthocyanin and carotenoid were compared using differential expression genes analysis. I do find the work of sufficient interest for publication on Horticulturae, after a revision. There are a few issues to address.

(1) The authors have not performed metabolomic analysis (refer to the definition of metabolomics), rather it’s more like a targeted analysis of anthocyanin and carotenoid. I suggest removing the term metabolomic from the title.

(2) Information for the two sweet cherry cultivars used in the current study needs to be clearly described. E.g., date of collection, how the identification of the plant cultivars being done, and who performed the identification.

(3) Over abundant of Figures in the manuscript, Fig. 1, 2 and 3 can be safely moved to Supplementary Information.

(4) Table 3; Can the authors explain why cyanidin-3-O-glucoside content in sample JNH S3 showing RSD of above 20%? Any associated errors during the measurement?

(5) Fig. 4, 5, and 7 are of poor quality, please improve to at least 300 dpi.

Reviewer 2 Report

The work is devoted to the study of the transcriptome and metabolome of two varieties of Sweet Cherry (Prunus avium L.)  at several stages of fruit ripening. The authors have done a large amount of analytical work and carefully analyzed the data obtained. The stated goal and objectives were fully disclosed by the authors.

I recommend the work for publication as presented.

As a small remark - the authors should point out the small changes they have made in the method of determination of anthocyanins - section 2.2 "Anthocyanin content was determined using the method described by Wei et al. (2015) with minor modification."

Also, the authors should note that the Supplementary Materials are missing in the presented version.

Author Response

small changes is 

The anthocyanins were extracted in 10 mL methanol (containing 0.05 % hydrochloric acid) for 24h at 4℃ in darkness. The anthocyanin content was measured by the TripleTOF 5600 LC-MS/MS (AB SCIEX, USA) at 520 nm.

Already described in the manuscript 2.2

Supplementary Materials has been modified

Reviewer 3 Report

Comments and Suggestions for horticulturae-2290377-peer-review-v1 to the authors

Transcriptomics in in crop research very common approach to identify gene difference and its regulation for different contrast traits.  The result of present research will be helpful for researchers to understand the transcriptomics and metabolomics of anthocyanin and carotenoid    in cherry as well as in other crops.   The manuscript was well written for most of the section except few section where improvement neededDiscussion section need more effort to explain outcome results with update literature/references. Manuscript requires minor revision with few recommended comments. 

Specific comments:

Comment 1: 

2.9. Statistical analysis 

Significant differences were calculated using one way analysis ………….and are shown in table with lower case…….samples. Authors need to mention the table number here in above text.

Comment 2.

Authors should present table 3 as a visualization image which will help to readers to easy understanding and comparative view in image results.

Comment3. Authors can add correlation (tables/plot) between anthocyanin and carotenoid at different stage.

Comment 4: Table 4 not explained in Results section. Authors should explain it in text for reader’s clarity. 

Comment5:  Discussion section need more efforts to improve it. It is surprised to see why authors did not use updated/current references to make discussion section strong? I strongly recommend using recent references to improve discussion section.

Author Response

Point 1: added table number

Point2:Due to the large amount of Table 3, it appears very crowded after being made into an image, so it is presented in the form of a table

Point3:Added during discussion

Point4:Table 4 is a measure of sequencing accuracy. As explained in 3.2, the data is reliable

Point5:Has been supplemented with recent references

Reviewer 4 Report

The present manuscript “Transcriptomic and Metabolomic Analysis of Anthocyanin and Carotenoid Biosynthesis in Red and Yellow Fruits of Sweet Cherry (Prunus avium L.) during Ripening” is devoted to the comparative analysis of genes involved in anthocyanin and carotenoid biosynthesis in fruits of sweet cherry in different ripening stages. Dtspite the fact that the pigments content and their biosynthesis have been studied in many plant species with different fruit colors, including sweet cherries, the authors obtained new data about the coloration change of ripening sweet cherry. They found that unigenes CHS and UFGT encoding anthocyanin biosynthesis were the most differently expressed genes between red and yellow varieties, but the expression pattern of carotenoid biosynthesis genes were relatively similar.

The methods used by the authors are clear described. The results obtained are statistically confirmed and well illustrated by the tables and figures.

The authors are well informed about studies in the field of their experiments that allowed them to identify novelty of their results.

There are several comments to the manuscript:

1)    Page 2 The “Koes and Verweij, 2005” is not included into the References List

2)    Page 6. There are no references to Table 1 and Table 2 in the text. Maybe the Table 1 is not necessary.

3)    Page 16. There is no year in the Reference No 6.

4)    Page 16. The Reference between 7 and 8 is not numbered

5)    Page 16. The Reference 10 seems to be published

6)    Page 17. it seems the References No 16 is not mentioned in the text

Author Response

All references have been modified

References No 16 is in 2.5

2.1 requires Table1 and Table2

Round 2

Reviewer 1 Report

With reference to the earlier reviewers’ comments, the authors have made some efforts to address some of these concerns in the current revised manuscript. However, the following comments have not been clearly addressed in the current version.  

**Please add details on how the identification of the plant cultivars being done, and who performed the identification. If plant material is identified or verified by Zhejiang Provincial Crop Variety Approval Committee, please indicate clearly. The word “approved” is not meaningful.

Author Response

Modify to ‘ plant material is identified  by Zhejiang Provincial Crop Variety Approval Committee.’

Through our team research We can't understand 'how the identification of the plant cultivars being done' "Because in China, the identification of new varieties is carried out by the Ministry of Agriculture of the People's Republic of China, I have not found any explanation for this in the relevant research papers."